# Proteomic Analysis of the Midgut Contents of Silkworm in the Pupal Stage

**DOI:** 10.3390/insects14120953

**Published:** 2023-12-15

**Authors:** Qinglang Wang, Lingzhen Yang, Tingting Tian, Yuanyuan Sun, Haonan Dong, Jing Gong, Yong Hou

**Affiliations:** Integrative Science Center of Germplasm Creation in Western China (CHONGQING) Science City, Southwest University, Chongqing 400715, China; 628wql@gmail.com (Q.W.); lzhyang1559@163.com (L.Y.); tingtingtian0502@163.com (T.T.); sunyuan_0416@163.com (Y.S.); haonandong.1995@gmail.com (H.D.); gongjing@swu.edu.cn (J.G.)

**Keywords:** *Bombyx mori*, midgut contents, metabolic activity, midgut degradation, trypsin, chymotrypsin

## Abstract

**Simple Summary:**

During the pupal stage, the midgut of *Bombyx mori* undergoes remarkable changes, condensing into the midgut lumen and, subsequently, undergoing decomposition. To comprehensively analyze this process of change, we utilized mass spectrometry to examine the midgut contents in the early and later stages of pupation. We detected a large number of enzymes involved in metabolism, proteases, inhibitors, chitin-binding proteins, chitinases, and storage proteins in the midgut contents during the pupal stage, indicating significant alterations within the pupal midgut lumen. Additionally, we have identified six proteases with potential gelatinolytic activity, which may be involved in the degradation of the midgut contents during the late stages of pupation. This research provides valuable insights into the mechanisms of tissue digestion and midgut remodeling in *Bombyx mori* pupae. The results serve as an important reference for comprehending the intricate processes taking place during this developmental stage in lepidopteran insects.

**Abstract:**

The silkworm *Bombyx mori*, a lepidopteran insect, possesses an 8–10-day pupal stage, during which significant changes occur in the midgut, where it first condenses into the yellow body, and then undergoes decomposition. To gain insights into this transformation process, proteomics was performed on *Bombyx mori* midgut contents on day 2 and day 7 after pupation. The results revealed the identification of 771 proteins with more than one unique peptide. An analysis using AgriGO demonstrated that these proteins were predominantly associated with catalytic activity. Among the identified proteins, a considerable number were found to be involved in carbohydrate metabolism, amino acid metabolism, lipid metabolism, nucleic acid degradation, and energy support. Additionally, variations in the levels of certain proteases were observed between the midgut contents on day 2 and day 7 after pupation. An in-depth analysis of the two-dimensional electrophoresis of the midgut contents on day 7 after pupation led to the identification of twelve protein spots with potential gelatinolytic activity. Among these, six proteases were identified through mass spectrometry, including the p37k protease, vitellin-degrading protease, chymotrypsin-2, etc. These proteases may be responsible for the digestion of the yellow body during the later stages of pupal development.

## 1. Introduction

The digestive tract consists of three distinct segments: the foregut, the midgut, and the hindgut. There are four different types of cells: the columnar cell, the Goblet cell, the stem cell, and the endocrine cell [1,2]. Columnar cells are the prevalent cell types with the ability to produce digestive enzymes. [3,4,5,6]. Goblet cells are commonly found in insects with alkaline gut cavities, primarily in Lepidopteran insects. They play a crucial role in alkalinizing the intestinal lumen and maintaining ion homeostasis in the midgut. [7]. Additionally, small stem cells located in the basal region of the epithelium [8,9] play a crucial role in midgut epithelial growth and tissue repair [10]. Moreover, a few scattered endocrine cells within the midgut epithelial cells regulate stem cell differentiation and enzyme secretion [11,12]. The midgut not only plays a vital role in the majority of digestive activities but also serves as a defense mechanism against pathogenic micro-organisms [9].

Interestingly, in certain insects, such as Thysanura, Blattella, Isoptera, etc., the midgut epithelium is completely renewed during each molt, whereas, in others like *Hippotion celerio*, Vanessa, etc., complete replacement occurs only at the pupal stage [12]. During the larval–pupal transition, the insect larval midgut epithelium undergoes programmed cell death under hormonal regulation [13,14]. The detached larval midgut epithelium is then pushed into the midgut lumen and transformed into an amorphous mass known as the yellow body [12]. The yellow body shows high apoptotic enzyme activity and an abundance of autophagy, indicating that the degradation of the larval epithelium is primarily accomplished through autophagy and apoptosis [13,14]. The autophagy is activated to promote tissue growth by facilitating lipolysis and providing nutritional support during the metamorphic process [15], while apoptosis is essential for tissue remodeling [16,17]. Autophagy is primarily determined by autophagy-related (ATG) proteins. Juvenile hormone (JH) can inhibit the expression of ATG proteins, thereby suppressing autophagy during metamorphic development and influencing midgut remodeling during this stage [18]. Through a genome-wide analysis of apoptosis-related genes in the silkworm, three caspases were identified, namely, BmCaspase-N, BmCaspase-1, and BmICE [19].

As the pupal stage progresses, larval cell membranes disintegrate, and their contents are released into the yellow body to serve as a source of energy and material for insect pupal development [20]. The activity of trehalase, sucrase, aminopeptidase N, and alkaline phosphatase significantly increases after pupation [13]. With insects not having access to a food source during the pupal stage, the yellow body serves as a crucial reservoir of energy and materials essential for their pupal development [21].

Apart from caspases and autophagy-related (ATG) genes, several proteases were reported to be related to insect midgut degeneration and remodeling [14]. For instance, a serine protease was isolated from the pupal yellow body of the *Sarcophaga peregrine*, showing a structural similarity to bovine trypsin and suggesting its role in disintegrating the larval midgut within the yellow body [22]. Another serine protease, p37k protease, was discovered in the silkworm midgut and exhibited gelatinolytic activity after pupation, indicating its potential involvement in degrading structural components within the gut lumen [23]. Additionally, SeCT34, a chymotrypsin in the larval midgut, has been speculated to participate in proteolytic remodeling during the larva–pupa transformation [24]. Cathepsins are a type of cysteine or aspartate protease that plays diverse roles in tissue remodeling during metamorphic development [25]. The expression of tissue proteases B, D, and L is induced by 20E, and their expression is associated with the degradation of larval midgut and fat body [26,27]. In *Delia radicum*, a cysteine protease is highly induced during the larval–pupal transition and thereafter. It is localized externally in the midgut tissue, suggesting the involvement of cysteine protease activity in tissue remodeling at the onset and during metamorphosis [28]. Furthermore, some evidence also indicates that tissue proteases play a triggering role in apoptosis. In the *Helicoverpa armigera* midgut, cathepsin L has the ability to cleave procaspase 1 to form mature caspase 1, activating cell apoptosis [29]. Knocking out cathepsin L in *Antheraea pernyi* results in the downregulation of some apoptotic gene expression and impairs fat body dissociation [30].

Despite the widely recognized role of the yellow body as a nutrient reservoir and studies reporting on the morphological and remodeling mechanisms of the midgut during metamorphosis, the understanding of the mechanisms primarily focuses on the degradation processes involving cell apoptosis and autophagy in the metamorphic development of the old midgut. There is still a lack of additional clues regarding the complete degradation mechanisms of the regressing midgut in the intestinal cavity during metamorphosis. The comprehensive analyses of its protein components, particularly during the pupa–adult stage, are relatively lacking. The contents of the silkworm midgut, especially, undergo a liquid–mass–liquid morphological process in our observation. In this study, we aimed to investigate the degradation process of the larval midgut at the pupal stage. We utilized proteomic methods to analyze the change process in the pupal midgut contents. Moreover, gelatin zymography, combined with mass spectrometry, was utilized to identify the proteases involved in the degradation of the larval midgut during the pupal stage. These data will serve as a valuable reference for understanding the changes in storage substances and nutrient support during the pupal and adult stages of insect development.

## 2. Materials and Methods

### 2.1. Biological Materials

The *Bombyx mori* strain D9L was provided by the Biological Science Research Center at Southwest University, China. The silkworms were reared on mulberry leaves under the conditions of temperature 25 ± 1 °C, photoperiod 12 h light/12 h dark, and 75% ± 5% relative humidity. To analyze the morphology of the midgut and its contents, pupae were dissected daily after pupation. The midguts were carefully placed on filter paper or 6-well cell culture plates with sterile saline and observed using a microscope (Cnoptec, Smart, Chongqing, China). The midgut contents were displayed through dissection. Afterward, the midgut and its contents were washed three times with sterile water, and each was then separately transferred into a 1.5 mL sterile centrifuge tube. All samples were immediately frozen using liquid nitrogen and stored at −80 °C for further analysis.

### 2.2. Protein Digestion and LC-MS/MS Analysis

The midgut content from day 2 and day 7 after pupation (P2 and P7) was prepared for analysis as follows: it was diluted with 8 M urea, and then subjected to centrifugation at 12,000× *g* for 10 min at 4 °C. The protein concentrations in the supernatant were determined using the Bradford method [31]. Subsequently, 200 μg of proteins were taken and treated by reducing them with 10 mM dithiothreitol and alkylating them with 50 mM iodoacetamide. The proteins were then digested with 4 μg of trypsin (1/50) in 50 mM NH_4_HCO_3_ at 37 °C for 24 h. The enzymatic peptides were collected by centrifugation at 12,000× *g* for 20 min. After desalination and lyophilization, the samples were resuspended in 0.1% formic acid and separated using the EASY-Spray column on the EASY-nLC 1000 system (ThermoFisher, Waltham, MA, USA). For peptide analysis, a mass spectrometer (ThermoFisher, Q Exactive, Waltham, MA, USA) was used in data-dependent mode. The instrument parameters for the MS2 and the full MS scans were set according to the previous method [32]. The mass spectrum resolution was adjusted to 70,000 for the full MS scan and 17,500 for the MS/MS scan. The maximum ion implantation times were 20 ms for the full scan, and the MS/MS scan time was set to 60 ms.

### 2.3. Two-Dimensional Electrophoresis

The 2-DE was conducted following a previously described method with slight modifications [33]. The midgut content of P7 was dissolved in an isoelectric focusing (IEF) sample buffer containing 1% Triton X-100 and 5% glycerol. After allowing it to stand at 4 °C for 40 min, the supernatant was collected by centrifugation at 12,000× *g* for 20 min at 4 °C. To remove small molecule substances, salts, or ions from the sample, dialysis tubing 100 (Njduly, retained molecular weight: 100 Da) was used in dialysate containing 1% Triton X-100 and 5% glycerol. A total of 200 µg of protein was re-suspended in 250 µL of rehydration buffer containing 1% Triton X-100, 5% glycerol, and 0.1% IPG buffer (Cytiva, pH 4–7, Marlborough, MA, USA). The immobilized dry IPG strips (Cytiva, 13 cm, pH 4–7, USA) were swelled overnight with rehydration buffer in Ettan IPGphor 3 (Cytiva, GE Ettan IPGphor 3, Marlborough, MA, USA). The IEF program was set as follows: 100 V for 2 h, 200 V for 1 h, 500 V for 1 h, 1000 V for 1 h, 5000 V for 1 h, and 5000 V for 10 h. Subsequently, the strips were equilibrated with equilibration buffer containing 0.375 M Tris-HCl pH 8.8, 2% SDS, 0.001% bromophenol blue, and 20% glycerol for 2 × 15 min, and then placed on 12% SDS-PAGE gels containing 0.1% of gelatin. The electrophoresis was carried out at 4 °C under 120 V until the dye reached the bottom. A parallel experiment was also performed on 12% SDS-PAGE gels without gelatin. Silver staining analyses were performed on gel without gelatin according to the previous method [34]. Three replicates were set for the 2-DE experiment reliability of the results.

### 2.4. Zymography Assay

After electrophoresis, zymography assays are performed to assess the gelatinolytic activity of proteases in the midgut contents. For 1-D zymography, different loading quantities of proteins (1, 4, and 8 μg) from the midgut content at P2 and P7 are separated using SDS-PAGE. For 2-D zymography, proteins from the midgut contents at P7 are separated using 2-DE. After electrophoresis, SDS in the gel was removed by a washing solution containing 50 mM Tris-base pH 7.4, 5 mM CaCl2, 1 μM ZnCl2, and 2.5% Triton X-100 for 40 min at 4 °C. Then, residual Triton X-100 is washed off three times for 20 min each using a rinse solution containing 50 mM Tris-base pH 7.4, 5 mM CaCl2, 1 μM ZnCl2, and 2.5% Triton X-100. The gels are then incubated for 48 h at 37 °C in a reaction buffer containing 50 mM Tris-base (pH 7.4), 5 mM CaCl2, 1 μM ZnCl2, and 0.02% Brij-35. During this incubation period, the proteases with gelatinolytic activity present in the gel will hydrolyze the gelatin, resulting in the appearance of clear bands or spots against a background of blue staining. After the incubation, the hydrolyzed gelatin bands or spots are visualized against a background of blue by staining with 0.2% Coomassie Blue R-250, 40% methanol, and 10% acetic acid, and destaining with 40% methanol and 10% acetic acid.

### 2.5. In-Gel Trypsin Digestion and LC-MS/MS Analysis

In 2-D zymography, we divided the gelatinolytic activity region into five zones. Around the I gelatinolytic activity zone, three protein spots were observed against a blue background, corresponding to silver-stained 2-DE. Using these three protein spots as references, along with their molecular weights, a total of 12 protein spots were identified and selected in silver-stained 2-DE. The silver-stained 2-DE spots that matched in the gelatinolytic area were carefully excised from the gel and collected in a 1.5 mL centrifuge tube. The proteins within these gel spots were subjected to in-gel reduction and alkylation. Subsequently, trypsin was used to digest the proteins, yielding peptides. To extract the peptides, 100 µL of extraction buffer containing 60% acetonitrile (ACN) and 5% formic acid was employed. The extracted peptides were then dried in a Speed Vac (ThermoFisher, Labconco CentriVap, USA) to eliminate the organic solvent. Following this preparation, the samples were separated and analyzed using LC-MS/MS, as previously described in Section 2.2.

### 2.6. Protein Identification, Quantification, and Analysis

The raw MS data were processed and analyzed using MaxQuant version 1.3.0.1. The search was performed on the silkbase (https://silkbase.ab.a.u-tokyo.ac.jp/cgi-bin/index.cgi) (accessed on 2 June 2022), which contains 16,880 protein sequences. To ensure the reliability of the results, proteins identified by MaxQuant as “potential contaminants” and “reverse” were excluded from further analysis. Proteins with a minimum of one unique peptide were retained. To compare the abundances of these identified proteins, iBAQ (intensity-based absolute protein quantification) was employed. Statistical analysis was carried out using a two-tailed Student *t*-test to assess significant differences between P2 and P7 content proteins. The threshold for determining significant differences was set at *p*-value < 0.05 and log2 fold-change >1 or <−1. The signal peptide of proteins was predicted using the online prediction website SignalP (http://www.cbs.dtu.dk/services/SignalP/) (accessed on 12 July 2022). For functional and gene ontology (GO) annotation of the identified proteins, the online website eggnog-mapper (http://eggnog-mapper.embl.de/) (accessed on 12 July 2022) was used. GO analysis of proteins was performed via Database for Annotation, Visualization and Integrated Discovery (DAVID, https://david.ncifcrf.gov/home.jsp) (accessed on 3 November 2022) to further characterize the biological functions of these proteins. Furthermore, functional enrichment analysis was conducted using AgriGO v2.0 (http://systemsbiology.cau.edu.cn/agriGOv2/) (accessed on 15 November 2022) to assign biological relevance to these proteins, visualizing through hierarchical tree graphs of GO terms. The heatmap was plotted by an online platform (https://www.bioinformatics.com.cn) (accessed on 16 November 2022) for data analysis and visualization.

### 2.7. Quantitative Real-Time PCR

Total RNA was extracted from the midgut (without contents) of *Bombyx mori* on day 2 after wandering (W2), the prepupal period (PP), and days 1 to 8 after pupation (P1 to P8) using the trizol reagent (Invitrogen, Carlsbad, CA, USA). We conducted three sets of replicated experiments, with three silkworm midguts sampled in each experimental group. To eliminate genomic DNA and synthesize cDNA, the PrimeScript™ RT reagent Kit (Takara, Tokyo, Japan) was used. The quantitative real-time PCR (qPCR), setting eukaryotic translation initiation factor 4A (sw22934) as an internal reference, was performed using the Applied Biosystems™ 7500 (ABI, USA) and SYBR Premix Ex Taq Kit (TaKaRa, Otsu, Japan). Thermal cycles were as follows: initial denaturation at 95 °C for 30 s, followed by 40 cycles of 95 °C for 3 s, and 60 °C for 30 s. The relative mRNA expression level was calculated using the 2^–∆∆Ct^ method. The specific primers used in the qPCR were listed in Appendix A. We used a two-tailed *t*-test and letter-marking method for statistical analysis of the samples.

## 3. Results

### 3.1. Morphological Changes of the Midgut in the Pupal Stage

We dissected and observed the midgut from day 1 to day 7 after pupation under the microscope. During the pupal stage, the midgut of the silkworm undergoes degeneration, and any residual food inside the midgut is expelled. From day 1 to day 4 after pupation, the midgut degenerates into an obovate shape surrounded by fat bodies, and the contents of the midgut appear as a homogenate with a significant amount of degenerated tissue (Figure 1a–f). On the fifth day after pupation, the midgut appeared as an oval shape, and the amorphous mass was observed filling the midgut lumen (Figure 1g,h). As the pupal stage progressed to day 6, the amorphous mass started to dissolve, and the midgut lumen was refilled with yellow fluid (Figure 1i,j). By the seventh day of pupation, the color of the midgut changed from yellow to blackish green, and the contents of the midgut began to decrease (Figure 1k,l).

### 3.2. Proteomic Analysis of Midgut Contents

To elucidate the changes in the midgut contents during the pupal stage, LC-MS/MS analysis was performed on the midgut contents of P2 and P7. After data cleansing and normalization, a total of 771 proteins were identified with more than one unique peptide, of which 386 proteins were identified in both P2 and P7 (Figure 2a). Furthermore, 551 proteins were annotated with at least one common GO term (Appendix A). Among the proteins present in both stages, 181 proteins exhibited statistically significant differences in abundance (*p* < 0.05) (Appendix A). Out of these, 159 proteins showed a log2(iBAQ p2/iBAQ p7) > 1, while 18 proteins had a log2(iBAQ p2/iBAQ p7) < −1 (Appendix A). In signal peptide prediction, 319 peptides have been predicted to possess the signal peptide (Appendix A).

### 3.3. Gene Ontology Analysis of Midgut Contents

GO function enrichment of the identified proteins was performed through the DAVID database. The results showed that the proteins identified on P2 were enriched in four biological processes, two cellular compounds, and twelve molecular function categories, whereas the proteins identified on P7 were enriched in one biological process, three cellular compounds, and eleven molecular function categories (Appendix A). The identified proteins were mainly clustered in molecular functional categories, with “catalytic activity” and “hydrolase activity” as the most enriched functions under the “molecular function” category (Figure 2b and Appendix A). The proportion of hydrolytic activity and catalytic activity was higher in P7 compared to P2 (Figure 2b). To determine the biological relevance of these midgut content proteins, singular enrichment analysis (SEA) from AgriGO was employed for functional enrichment analysis. In the molecular function enrichment analysis, proteins were found to be significantly enriched in hydrolase activity (4.53 × 10^−16^) under the catalytic activity (9.07 × 10^−6^), of which the peptidase activity, serine hydrolase activity, and hydrolyzing O-glycosyl compounds reached 4.53 × 10^−16^, 5.54 × 10^−9^, and 1.25 × 10^−9^ (Figure 3a), respectively, implying an important role for these proteins in protein degradation in the pupal midgut contents. For the cellular component analysis, the proteins were found to be significantly enriched in the extracellular region (2.85 × 10^−7^, Figure 3b). In terms of biological processes, the proteins were mainly enriched in the oxidation–reduction (4.74 × 10^−12^), proteolysis (3.5 × 10^−10^), and carbohydrate metabolic process (1.78 × 10^−17^, Figure 3c) under the metabolic process category.

### 3.4. Proteins Involved in Degradation Processes

A GO enrichment analysis indicated that the proteins in the midgut content were predominantly enriched in catalytic activity, hydrolytic activity, and various metabolic processes. This suggests the occurrence of protein degradation and material metabolism activities within the midgut contents. Therefore, the metabolic enzymes and proteases identified from the midgut content was further investigated. In the midgut contents, a total of 161 enzymes involved in metabolism were identified. A significant number of enzymes were associated with carbohydrate metabolism (67 proteins) and amino acid metabolism (36 proteins), and 12 proteins were related to phosphate metabolism, 18 proteins were involved in nucleic acid metabolism, and 14 proteins participated in lipid metabolism (Appendix A). A total of 93 proteases were identified, including 41 serine proteases, 28 carboxypeptidases, 16 aminopeptidases, 2 cysteine proteases, 2 metalloproteinases, 2 peptidases, and 2 cathepsins (Appendix A). The peritrophic membrane is primarily composed of cuticular proteins and chitin. During the metamorphic development of the silkworm, the peritrophic membrane undergoes degradation and is gradually expelled from the body. In this study, a variety of chitin-related proteins were identified in the midgut contents, including 24 epidermal proteins, 5 chitinases, and 2 beta-N-acetylglucosaminidase (Appendix A). Meanwhile, some nutritional proteins, along with 30 kDa lipoproteins possessing energy and material storage capacity, were also identified in the midgut contents (Appendix A).

### 3.5. Gelatinolytic Activity of Midgut Contents

During the pupal stage, the contents of the midgut undergo a liquid–mass–liquid transition (Figure 1). This dynamic change is indicative of significant remodeling and degradation processes occurring in the midgut. The proteomic data revealed that numerous proteases were highly expressed in the late pupal stage, suggesting their crucial role in the degradation of the midgut contents. To further investigate the activity of tissue-degrading enzymes, a gelatin zymography assay was performed. The results showed that the gelatinolytic activity of P7 was observed to be stronger than that of P2, particularly around molecular weights of 25 kDa and 15 kDa (Figure 4). To identify the specific proteins responsible for this tissue-degrading activity, a 2D-zymography analysis was conducted for P7. The results revealed that the strongest gelatinolytic activity was localized in five distinct areas around 35–50 kDa at pH 5–6 and 15–25 kDa at pH 4–5 (Figure 5a).

### 3.6. Identification of Potential Gelatinolytic Proteins

Due to containing gelatin fragments, the identification of proteins becomes more difficult in 2D zymography. Therefore, a conventional SDS-PAGE gel was set in parallel without gelatin. In the 2D silver-stained gel, twelve spots that aligned with the gelatinolytic activity area were carefully extracted and subjected to mass spectrometry analysis (Figure 5b). The analysis revealed the identification of a dozen proteins from each spot, resulting in a total of 112 proteins across these twelve spots (Appendix A). As the focus was on identifying proteases among these proteins, only proteins with gelatinolytic activity were selected for further analysis, including XP_037871589.1, XP_028037175.1, XP_021205514.2, XP_028038802.1, NP_001128675.1, and XP_028033074.1 (Table 1). Some of these proteases appeared in multiple spots on the silver-stained gel, indicating either their high expression level in P7 or post-translational modifications occurring in these proteins.

### 3.7. Quantitative Real-Time PCR

To further validate the proteomic data obtained from mass spectrometry and zymography, we conducted a qPCR analysis to investigate the expression of six protease genes on the midgut from the wandering stage to the late pupal stage. The results indicated that XP_028033074.1 exhibits expression in both the early pupal and late pupal stages (Figure 6a). Additionally, NP_001128675.1, XP_028038802.1, and XP_028037175.1 exhibited high expression levels on the second day of the wandering stage (Figure 6b–d). In contrast, XP_021205514.2 and XP_037871589.1 showed minimal expression in the early pupal stage but demonstrated significantly elevated expression levels during the later pupal stage (Figure 6e–f). These findings align with the mass spectrometry data presented in Section 3.2 (Figure 6, Table 1 and Appendix A).

## 4. Discussion

The midgut of lepidopteran insects serves as an important tool for studying intestinal morphogenesis and differentiation in insects [13,35,36]. The silkworm *Bombyx mori*, as a representative of Lepidoptera, holds great significance not only due to its economic value but also as a promising model for exploring insect genetics, development, and immunity [37]. During the larval–pupal stage, the midgut of the silkworm undergoes drastic transformations. Studies on the ultrastructure and cytochemistry of the silkworm midgut have been performed to observe its morphological changes during degeneration and remodeling [14,20,38,39]. The midgut epithelium of the silkworm exhibits the remarkable ability to recycle molecules derived from the yellow body [20]. Therefore, investigating the degeneration of the yellow body during the pupal stage can provide valuable insights into understanding the developmental mechanism of adult midgut stem cells. The midgut was observed to contain a compact mass with a thin epithelium during the middle stage of the pupa (Figure 1). Although several studies have previously been reported on insect midgut morphology [21,36,40], we remain uncertain about the structural differences in the midgut between the silkworm and other lepidopteran insects during the pupal and adult stages, especially considering that the silkworm does not feed. Hence, a further investigation and comparison of the cytology and morphology between *Bombyx mori* and other lepidopteran insects are warranted.

### 4.1. Proteins Involved in Metabolism

In the pupal stage of insects, previous studies [13,20] have speculated that nuclear debris, peptides, lipids, glycogen, and free sugars are released due to damage to the plasma membrane. In this study, we identified a total of 161 proteins involved in various metabolic processes, including 67 related to carbohydrate metabolism, 36 involved in amino acid metabolism, 14 associated with lipid metabolism, 18 participating in nucleic acid metabolism, etc. (Appendix A). Phosphates participate in metabolic processes and many reactions of phosphates are involved in energy metabolism [41,42]. Our dates indicate the involvement of twelve phosphate-related proteins in the midgut content, pointing towards active energy conversion during the pupal stage (Appendix A).

There were 95 metabolism-related proteins exhibiting significantly elevated levels at P2 (Appendix A, Figure 7a), such as ATP synthase, lipase, and aminoacylase, suggesting the presence of vigorous metabolic activities. After autophagy or apoptosis, the contents inside the cells enter the midgut lumen, which includes various proteins and enzymes [14]. These metabolic enzymes may originate from those cell contents, and further participate in the digestion of nuclear debris, peptides, lipids, glycogen in the midgut lumen to promote the recycling and utilization of substances. Meanwhile, some proteins related to metabolism showed higher expression levels during the late pupal stage. For instance, apyrase-like (XP_004927708.1, Appendix A; Figure 7a), an energy-related enzyme known to hydrolyze ATP or ADP to release energy [43,44], was more prominent during this stage. This likely indicates that, after a prolonged fasting period during the pupal stage, apyrase is required to provide energy for the development from a pupa to an adult. β-Galactosidase, an exoglycosidase, is involved in the hydrolysis of lactose, and the degradation of glycoprotein. It was universally expressed in various organisms and plays an important role in biological metabolism [45,46]. β-Galactosidase was expressed at multiple tissues including the midgut of the silkworm during the pupal stage [47]. The content of β-Galactosidase (XP_037869036.1) was only identified in day 7 after pupation, suggesting its involvement in carbohydrate metabolism during the transformation from a pupa to an adult.

### 4.2. Proteases and Inhibitors

As the main digestive organ, the midgut possesses a strong capacity to secrete various digestive enzymes. In this study, a total of 93 proteases were identified from the pupal midgut content, including 41 serine proteases, 16 aminopeptidases, 28 carboxypeptidases, etc. (Appendix A). Previous omics-based studies have analyzed the possible presence of proteases in the midgut [48,49,50]. In 2020, Jiang et al. performed transcriptomic and proteomic analyses to identify factors responsible for digestion, of which 62 serine proteases, 10 aminopeptidases, and 10 carboxypeptidases were identified, respectively [48]. Among them, 15 proteases were present in our omics data, mainly consisting of aminopeptidases and carboxypeptidases (Appendix A). One of the aminopeptidases, Aminopeptidase N (P_KWMTBOMO04889, Appendix A), showed elevated levels in the midgut content of P7. Aminopeptidase N, as a representative involved in digestion, was investigated and showed significant activity in the silkworm midgut brush border [20]. During the pupal stage, the protein mixture has been degraded to peptides. Therefore, these exopeptidases were responsible for producing single amino acids to make them available for absorption [20,51]. Serine protease is a type of protein hydrolase with a conserved His-Asp-Ser catalytic triad active site domain, participating in processes such as insect digestion, development, and immune response. Experiments involving RNA interference (RNAi) indicate that LsCLIP3, a Clip-domain serine protease, on *Lasioderma serricorne* plays a crucial role in the molting and immune responses of larvae to pupae [52]. Drosophila cSPH35 and cSPH242 are involved in mediating the activation process of PPO1, and the activation of PPO1 is essential for early melanization in adult [53]. In Drosophila, the stubble protease can degrade the apical matrix, inducing the elongation of wings and legs [54]. In our study, various serine proteases, including Clip-domain serine protease, serine proteinase stubble, and SPH, were detected in the intestinal contents during the pupal stage (Appendix A). These proteases may play diverse roles and are of significance in the degradation of midgut and immune processes in pupae. Furthermore, molting fluid carboxypeptidase A was also detected in the midgut contents during the pupal stage (NP_001036933.1, Appendix A). This protease has been reported to be present in the molting fluid at *Bombyx mori* during the larval–pupal transition, participating in the molting process by degrading proteins in the old cuticle layer and contributing to the recycling of amino acids [55]. In summary, various proteases participate in the degradation of degenerated tissues in the silkworm pupal midgut. Similarly, a variety of proteases, including serine protease, peptidase, and carboxypeptidase, were detected in the molting fluid of silkworm from larvae to pupae and pupae to adult [56]. These proteases play a role in the further digestion of degenerated tissues or cells, providing conditions for subsequent pupal development.

In addition to proteases, over 31 protease inhibitors were identified during the two periods of our study. Some of these inhibitors exhibited different expression levels at different stages. Most inhibitors were highly expressed in the midgut contents of P2, while two inhibitors showed high expression levels in the late pupal midgut contents (Figure 7b, Appendix A). A significant increase in carboxypeptidases was observed in the late pupal stage, accompanied by an increase in inhibitors (Figure 7b, Appendix A). This suggests that these inhibitors may play a role in regulating enzyme activity during the late pupal stage.

Furthermore, protease inhibitors were also assumed to be involved in immunity. *Pathogenic fungi* can invade insects through enzymatic degradation [57,58]. A TIL-type serine protease inhibitor, BmSPI38 (XP_004924399.1, Appendix A), has been found to restrict the conidium germination and penetration process of the *Beauveria bassiana* by inhibiting cuticle-degrading proteases [59,60]. Additionally, BmSPI11 (XP_037867815.1, Appendix A) and BmSPI6 (XP_012547703.1, Appendix A) were also identified, which was induced by infection with pathogenic micro-organisms [61]. These protease inhibitors may be involved in the defense against pathogen invasion during midgut remodeling.

### 4.3. Chitin-Related Proteins and Nutritional Proteins

The insect midgut epithelium is protected by a physical barrier, named the peritrophic matrix (PM), which consists of an organized lattice containing chitin and chitin-binding proteins [62]. During the larval–pupal metamorphosis, the peritrophic membrane undergoes degradation [12,63]. A high abundance of peritrophins including chondroitin proteoglycan-2, cuticular protein RR-1 motif 24, peritrophin 1, and CPAP3-type protein were identified in the midgut content (Appendix A). Additionally, numerous chitinases and beta-N-acetylglucosaminidases were identified in our data, responsible for digesting chitin [64,65,66]. It is worth noting that several R&R-type cuticular proteins were also present in the midgut content of P2. R&R-type cuticular proteins also have a chitin-binding domain, which is typically expressed in the epidermis but not found in the peritrophic matrix [67]. Based on transcriptome-based expression analyses of the midgut [68], we assume these R&R-type proteins originate from midgut epithelial cells.

Many nutritional proteins such as the storage protein and 30K protein were identified in the midgut contents of P2 but were nearly absent in the midgut contents of P7 (Figure 7c, Appendix A), which also suggests that pupae gradually degrade these nutrients during development. However, a 30K protein named Bmlp2 (WP_149822408.1, Figure 7c) exhibited high expression in the late pupal stage. A transcriptional analysis of Bmlp2 also indicates a different expression pattern compared to other 30K genes, showing continuous expression until adulthood [69]. These results indicated that different 30K proteins may be utilized in a specific order to supply long-term storage nutrients during the pupal–adult stage.

### 4.4. Proteinase by Gelatin-Zymography

Gelatin zymography is a valuable tool for studying hydrolytic abilities [70]. In vertebrates, gelatin-degrading activity is an essential marker for matrix metalloproteinase 2 (MMP-2, gelatinase A) and matrix metalloproteinase 9 (MMP-9, gelatinase B). These enzymes play a crucial role in degrading various extracellular matrices (ECMs) and are thus significant in biological and pathological studies [70,71]. In *Drosophila melanogaster* and *Bombyx mori* fat bodies, the knockout of MMPs significantly affects the dissociation and development of tissues [72,73]. Notably, in our study, only a small amount of MMP was detected within P2 midgut contents. However, we identified several proteinases with potential gelatinolytic activity, most of which are serine proteases, including trypsins and chymotrypsins.

A serine protease named p37k protease, with a molecular mass of 37 kDa, is associated with the metamorphic remodeling of the midgut and exhibits significant gelatinolytic activity after pupation [23]. The p37k protease was identified in several spots on the 2D gel (Table 1) and exhibited strong gelatinolytic activity in the pupal midgut content. Our previous research shows that p37k not only functions in the midgut but also plays an essential role in the degradation process of epidermal protein [74].

Interestingly, we identified two small-molecular-weight proteins with gelatin hydrolysis activity, namely, vitellin-degrading protease precursor (XP_021205514.2, Table 1) and chymotrypsin-2-like (XP_037871589.1, Table 1). Quantitative PCR results also demonstrated their later expression in pupae (Figure 6). Vitellin-degrading protease is a trypsin-like serine protease purified from silkworm eggs [75,76]. Its transcriptional activity was found to occur at the head pigmentation stage [75,76]. Moreover, an analysis of over 50 cDNA libraries from *Manduca sexta* revealed that serine proteases in group-11 transcribe at higher levels in the midgut during the pupal and adult stages [48,77]. The vitellin-degrading protease gene is an ortholog of SP251, belonging to group-11 serine proteases of *Manduca sexta*.

Most insect chymotrypsins are reported to be involved in food digestion in the insect midgut. For instance, in *Stomoxys calcitrans*, two chymotrypsins expressed predominantly in the opaque region were considered to be involved in digestion [78]. Similarly, both Slctlp1 and Slctlp2 of *Spodoptera litura* were downregulated by starvation, whereas they were upregulated again upon re-feeding [79,80]. However, some chymotrypsins have been found to be associated with molting. TcCTLP-5C and TcCTLP-6E from *Tribolium castaneum* exist in the carcass, are secreted into the molting fluid, and contribute to insect molting [81]. Here, we discovered that chymotrypsins may serve another function, participating in the hydrolysis of tissues to supply long-term nutrients for the development of pupae and adults.

## 5. Conclusions

In conclusion, this study provided valuable insights into the pupal midgut content, identifying numerous proteins involved in sugar metabolism, amino acid metabolism, lipid metabolism, nucleic acid degradation, and energy support. The combination of proteomics data and gelatin activity experiments allowed for the identification of a significant number of proteases present in the midgut during various developmental stages. These results not only offer additional targets for studying midgut degeneration and remodeling but also contribute to a better understanding of the role of proteases during the pupal–adult developmental stage.

## Figures and Tables

**Figure 1 insects-14-00953-f001:**
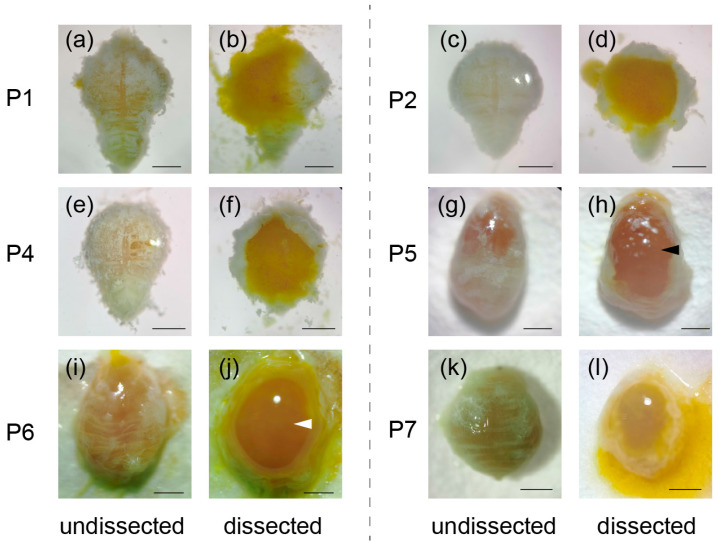
Midgut and contents morphology in the pupal stage. The midgut was observed on different days after pupation: day 1 (**a**,**b**), day 2 (**c**,**d**), day 4 (**e**,**f**), day 5 (**g**,**h**), day 6 (**i**,**j**), and day 7 (**k**,**l**). The black lines indicate the scale. Bars: (**a**–**l**) 2 mm; P1: day 1 after pupation, P2: day 2 after pupation, P4: day 4 after pupation, P5: day 5 after pupation, P6: day 6 after pupation, and P7: day 7 after pupation. Black arrow: solidified midgut contents; White arrow: midgut contents re-dissolved; Undissected: observing the pupal midgut without dissecting the midgut epithelium; Dissected: dissecting the midgut epithelium to observe the pupal midgut and midgut contents.

**Figure 2 insects-14-00953-f002:**
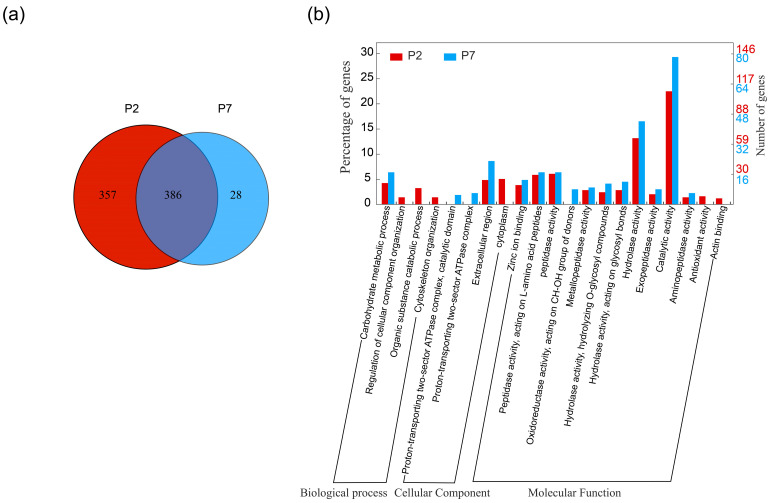
Protein identification and analysis in the midgut contents. Venn diagram analysis of proteins in the midgut contents (**a**). A comparison was made between the number of proteins identified in P2 and P7. A total of 386 proteins were found to be identified in both P2 and P7. Gene ontology (GO) classification of the midgut contents (**b**). Proteins identified from the midgut contents were classified into “cellular component” and “molecular function” categories using DAVID based on the GO terms. The number of proteins mapped to the GO terms for P7 is represented in blue, and, for P2, in red, as shown in the right panel. P2: day 2 after pupation, and P7: day 7 after pupation.

**Figure 3 insects-14-00953-f003:**
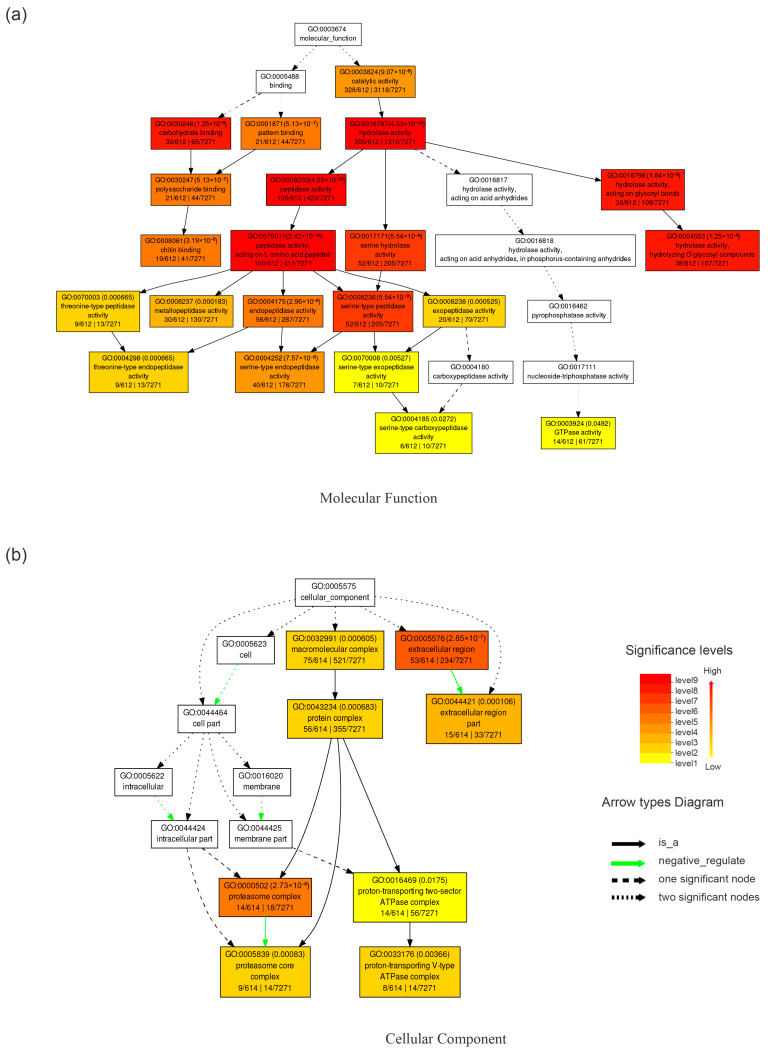
Hierarchical tree graph of GO terms for proteins in the midgut contents. The hierarchical tree graphs were constructed using AgriGO v2.0. In each graph, each box represents a GO term and is labeled with its term definition, GO number, and statistical information. The significant terms with adjusted *p* ≤ 0.05 are colored. The GO categories are as follows: (**a**) molecular function, (**b**) cellular component, and (**c**) biological process.

**Figure 4 insects-14-00953-f004:**
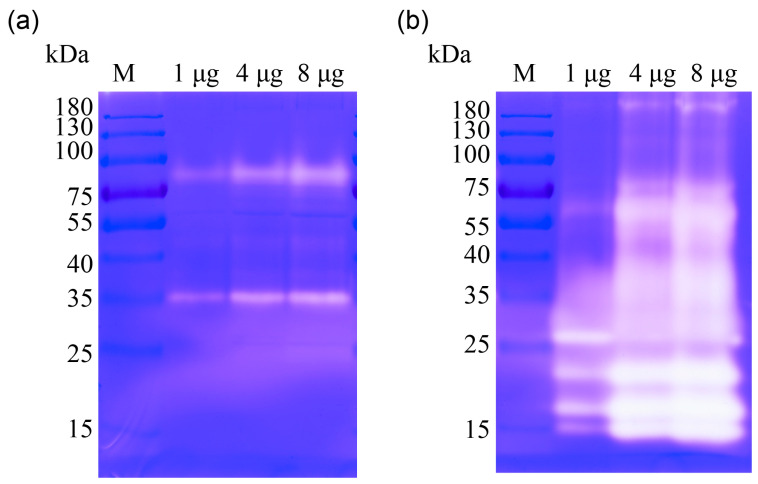
Analysis of zymography in the midgut contents of the pupal stage. The gels were incubated at 37 °C in a reaction buffer for 42 h to detect proteolytic activities. The gelatinolytic activities of midgut contents on day 2 after pupation (**a**) and day 7 after pupation (**b**) were detected in 10 μL loading volumes of each lane, containing 1, 4, and 8 μg of protein, respectively. The gelatinolytic bands appeared as white bands against a blue-stained background staining by Coomassie blue. M: marker.

**Figure 5 insects-14-00953-f005:**
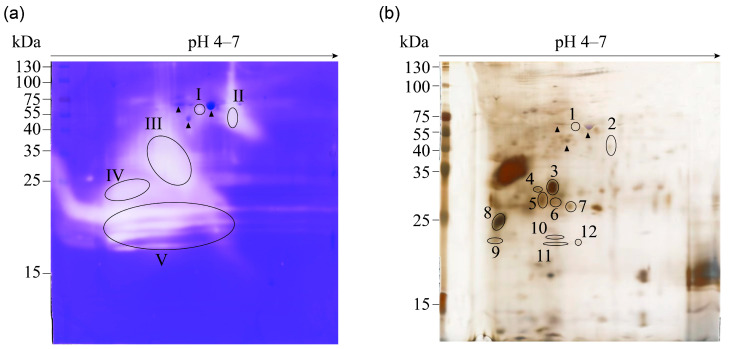
2–DE gelatin zymogram and 2-DE pattern of midgut contents. The midgut contents (200 µg) were separated on 13 cm IPG dry strips (pH 4–7) and, subsequently, subjected to 12% SDS–PAGE under non-reducing conditions. (**a**) The 2–DE gelatin zymography, with the gels copolymerized with 0.1% gelatin. The proteolytic activities were visualized as five clear zones ((**a**), I–V) against a blue-stained background staining by Coomassie blue. (**b**) 2–DE was performed on gels without gelatin staining but rather with silver staining. A total of 12 spots ((**b**), 1–12) corresponding to the clear zones observed in the gelatin zymogram were identified in the 2D silver-stained gel. Black triangle: reference protein spots.

**Figure 6 insects-14-00953-f006:**
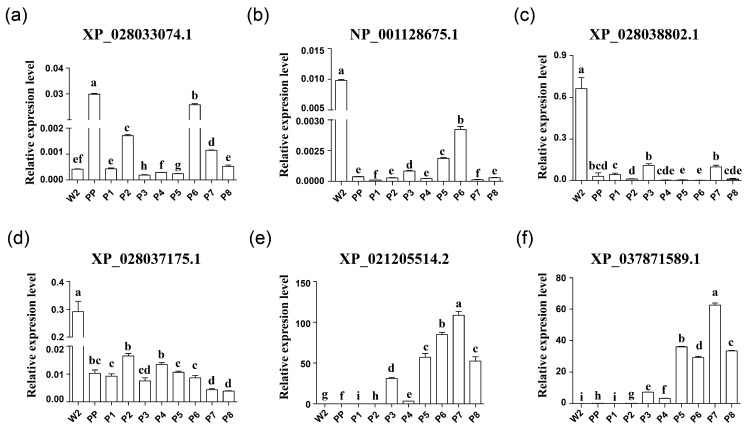
Pupal expression pattern of proteases identified from 2-DE gelatin zymogram as detected by qRT-PCR. (**a**–**f**) The expression patterns of neutral endopeptidase (XP_028033074.1), 37 kDa protease precursor (NP_001128675.1), putative peptidase (XP_028038802.1), chymotrypsin-2 (XP_028037175.1), vitellin-degrading protease (XP_021205514.2), and chymotrypsin-2-like (XP_037871589.1) during the pupal stage. The midgut without contents was collected at different time points: W2 (day 2 after wandering), prepupal period (PP), and P1 to P8 (days 1 to 8 after pupation). The eukaryotic translation initiation factor 4A (silkworm microarray probe ID: sw22934) was used as an internal control. Letter-marking method: Datasets with the same letter or sets containing identical letters indicate no significant difference, while different letters indicate a significant difference between the two datasets. Significance level is set at 0.05, and lowercase letter markings are used.

**Figure 7 insects-14-00953-f007:**
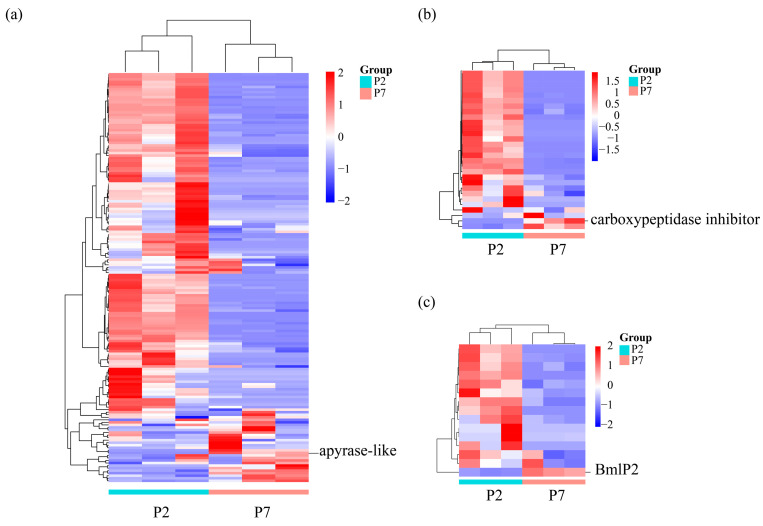
Expression profiles of metabolic enzymes, protease inhibitors, and low-molecular-weight lipoprotein. (**a**) Heatmap of metabolic enzymes in P2 and P7. (**b**) Heatmap of protease inhibitors in P2 and P7. (**c**) Heatmap of low-molecular-weight lipoprotein in P2 and P7. P2: day 2 after pupation. P7: day 7 after pupation. apyrase-like: XP_004927708.1, carboxypeptidase inhibitor: XP_028035896.1, Bmlp2: WP_149822408.1.

**Table 1 insects-14-00953-t001:** Identification of potential gelatinolytic proteins.

Spot	Protein ID	Silkbase ID	Description	MV (kDa)	GO Names List	Avg. iBAQ p2	Avg. iBAQ p7
Spot 2	XP_028038802.1	KWMTBOMO12335	Putative peptidase	77.5	P: proteolysis F: cysteine-type peptidase activity	1.00 × 10^9^	2.78 × 10^6^
	XP_028033074.1	KWMTBOMO00333	Neutral endopeptidase	63.2	F: metalloendopeptidase activity P: proteolysis	9.57 × 10^7^	3.47 × 10^7^
Spot 3 and 4, 5	XP_028037175.1	KWMTBOMO10720	chymotrypsin-2	31.0	F: serine-type endopeptidase activity P: proteolysis	1.07 × 10^7^	2.50 × 10^6^
	NP_001128675.1	KWMTBOMO03608	37 kDa protease precursor	30.3	F: serine-type endopeptidase activity P: proteolysis	3.91 × 10^7^	1.62 × 10^7^
Spot 8	XP_021205514.2	KWMTBOMO16414	Vitellin-degrading protease	27.2	F: serine-type endopeptidase activity P: proteolysis	0	3.81 × 10^8^
Spot 6 and 7, 8, 9, 10	XP_037871589.1	KWMTBOMO08714	chymotrypsin-2-like	25.1	F: serine-type endopeptidase activity P: proteolysis	0	4.33 × 10^8^

“P7” The unique proteins that were only identified on day 7 after pupation.

## Data Availability

The data are available upon request.

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
