# Peer review of "Proteomic Analysis of the Midgut Contents of Silkworm in the Pupal Stage"

_insects, 2023, doi:10.3390/insects14120953_

Round 1
Reviewer 1 Report
Comments and Suggestions for Authors
This manuscript by Wang et al. is a fairly straightforward proteomics analysis of the midgut proteome during metamorphosis in Bombyx mori. The authors examined two time points during early pupation (2 days post-pupation) and late pupation (7 days). During these times the larval midgut is breaking down and provides nutrients to non-feeding pupae, some of which will be used to reconstitute the adult midgut. The authors identified over 700 proteins, many of which would be expected, including digestive enzymes, apoptotic proteins, metabolic proteins, etc that are used to break down this larval tissue. The authors also used 2-DE gelatin zymograms to identify a range of proteins that are involved in active digestion and breakdown of the gut. Six of these were found to be upregulated at the 7 day time point and qPCR was utilized to validate transcript expression during pupation. Overall this work is interesting and sheds some light on an understudied tissue rearrangement. Although the focus on the midgut only during pupation does limit the usefulness of these data a bit. Overall, I think this manuscript will be of interest to the readers of Insects and I have only a few minor comments.
My greatest concern is the qPCR data for which the authors need to provide additional clarification and details. 1) How many times were the qPCR data replicated with unique biological samples (i.e., not technical replicates). 2) What were the qPCR conditions (temps, cycles, etc)? 3) What were the qPCR primer efficiencies? 4) There is no indication of which samples were significantly different from others? Was any statistical analysis performed? If not this needs to be conducted.
Minor points:
Figures 2 and 3 are not legible. The font is far too small to read, especially for 3.
In figure 5 how were the spots on the silver stained gel selected? Most of the gelatin staining is a broad, white smear and it is not clear how specific spots were chosen over others as being responsible for the gelatin digestion.
Although it is mentioned in the methods, the authors should clarify that the qPCR was performed on midguts, not whole bodies, in both the results and figure legend.
Comments on the Quality of English Language
Some edits to the quality of the English is needed, but it is generally acceptable.
Author Response
Dear Reviewer:
Thank you for taking time out of your busy schedule to review the manuscript. Now we have carefully corrected and responded to your comments point by point. All modified sections of the article are highlighted in blue font. The revision is described in the attachment. Please see the attachment.
Sincerely,
Yong Hou

Reviewer 2 Report
Comments and Suggestions for Authors
In this manuscript, Wang et al. presents a proteomic analysis of the midgut contents of the silkworm Bombyx mori during the pupal stage. They performed LC-MS/MS proteomics on midgut contents at day 2 and day 7 after pupation, and identified 771 proteins, many of which were involved in metabolism, proteolysis, and nutrient storage/degradation. The authors also conducted two-dimensional electrophoresis and gelatin zymography assays to identify proteases responsible for midgut degradation, leading to the identification of several serine proteases and metalloproteases. Overall, this is a fundamental analysis providing insight into the dynamic changes occurring in the silkworm midgut during metamorphosis, although it lacks sufficient functional interpretation of the data. There are some issues that should be addressed:
Major comments:
1. The introduction provides inadequate background on the current knowledge about insect midgut remodeling during metamorphosis. The authors should expand the introduction to provide more context, cite recent relevant studies, and clearly state the gap in knowledge this study aims to address.
2. The manuscript lacks a clear hypothesis and the proteomics appears largely exploratory. The authors should clarify the specific research objectives and questions guiding this study in the manuscript.
3. The results are not presented in a logical flow and many findings are not sufficiently explained. For example, the identified metabolic proteins are listed but their biological significance is not discussed. The authors should thoroughly explain all major findings.
4. The functional analysis of the proteomic data is very superficial and lacks statistical assessment. Using tools like DAVID or Panther to identify enriched pathways/functions, along with statistical enrichment scores, would strengthen this analysis.
Minor comments:
Figure quality is poor – some panels are too small in Fig.3, resolution is low. Should optimize figure size and quality.
Provide full gene/protein IDs in figures and tables, not just SilkBase numbers.
Author Response

(The authors gave the same response as above.)

Round 2
Reviewer 2 Report
Comments and Suggestions for Authors
The authors have been very responsive to the original critiques, and the manuscript is now much stronger. I have no additional remarks.
Author Response
Dear Reviewer:
Thank you for taking time out of your busy schedule to review the manuscript and provide constructive comments, which have improved the quality of our manuscript.
Sincerely,
Yong Hou